# Closed-Loop Temperature and Force Control of Additive Friction Stir Deposition

**Glen R. Merritt** [1] , **Malcolm B. Williams** [1] , **Paul G. Allison** [1,2] , **James B. Jordon** [1,2] , **Timothy W. Rushing** [3] and **Christian A. Cousin** [1,*]

1    Department of Mechanical Engineering, The University of Alabama, Tuscaloosa, AL 35487, USA
2    Manufacturing at the Point-of-Need Center (MPNC), The University of Alabama, Tuscaloosa, AL 35487, USA
3    US Army ERDC, Vicksburg, MS 39180, USA
*    Correspondence: cacousin@eng.ua.edu

**Abstract:** Additive Friction Stir Deposition (AFSD) is a recent innovation in non-beam-based metal additive manufacturing that achieves layer-by-layer deposition while avoiding the solid-to-liquid phase transformation. AFSD presents numerous benefits over other forms of fusion-based additive manufacturing, such as high-strength mechanical bonding, joining of dissimilar alloys, and high deposition rates. To improve, automate, and ensure the quality, uniformity, and consistency of the AFSD process, it is necessary to control the temperature at the interaction zone and the force applied to the consumable feedstock during deposition. In this paper, real-time temperature and force feedback are achieved by embedding thermocouples into the nonconsumable machine tool-shoulder and estimating the applied force from the motor current of the linear actuator driving the feedstock. Subsequently, temperature and force controllers are developed for the AFSD process, ensuring that the temperature at the interaction zone and the force applied to the feedstock track desired command values. The temperature and force controllers were evaluated separately and together on setpoints and time-varying trajectories. For combined temperature and force control with setpoints selected at a temperature of 420 °C and a force of 2669 N, the average temperature and force tracking errors are $5.4 \pm 6.5$ °C ($1.4 \pm 1.6\%$) and $140.1 \pm 213.5$ N ($5.2 \pm 8.0\%$), respectively.

**Keywords:** Additive Friction Stir Deposition (AFSD); temperature control; force control; closed-loop control

## 1. Introduction

Friction stir welding (FSW), first introduced in 1991 [1], has several advantages over traditional fusion-based welding practices. Because melting is not necessary to achieve metallurgical bonding in FSW, the material remains in the solid-state thus allowing a reduction in the size of the heat-affected zone, yielding stronger material bonds [2]. Additionally, FSW does not rely on metallic bonding of the materials because it creates a mechanical bond between the crystalline structures of the joined metals [3]. Using friction [4,5] to heat the material to 60–90% of the melting temperature allows materials to flow and create metallurgical bonds without reaching a liquid state [3,6]. Non-beam-based processes are beneficial because they require low energy and produce low material waste during bonding or repair [7].

Analytical and conceptual models associated with FSW [8,9] demonstrate some of the inherent difficulties when controlling the FSW process [10]. For example, convective heat transfer is a strongly nonlinear phenomenon subject to the difference in process and ambient temperature [11], and material properties are diificult to model (e.g., the elastic modulus of a material is temperature-dependent). Due to the complexity, atomistic simulation and Monte Carlo methods have been used to investigate the FSW process [12,13]. To the best of the authors' knowledge, limited research focuses on changing mechanical properties of common materials (e.g., 6061 aluminum alloy) at temperatures above half the melting

temperature and most research tends to focus on temperatures at which the mechanical properties degrade [14]. To best model solid-state welding, material properties such as the temperature-dependent elastic modulus and plasticity should be investigated because they affect the process dynamics and behaviors.

Another solid-state manufacturing process closely related to FSW is Additive Friction Stir Deposition (AFSD) [15–18], while similar, AFSD is distinguished from FSW because AFSD uses a consumable feedstock within a hollow nonconsumable rotating tool. The feedstock is forced through the tool and deposited onto a build plate to form a structure as shown in Figure 1. Like FSW, AFSD uses frictional heat generation to induce severe plastic deformation in the material (i.e., the feedstock) without melting and subsequently to achieve mechanical material flow to deposit the feedstock in a layer-by-layer process, analogous to other additive manufacturing processes [19–22]. AFSD is beneficial in comparison to other additive manufacturing processes because it requires lower energy compared to other manufacturing processes such as laser sintering [23]. Additionally, a wide variety of materials—such as aluminum alloys [24–28], magnesium alloys [29,30], copper [31,32], Inconel [33,34], and titanium alloys [35]—can be used in AFSD. While modeling and simulation of AFSD focus on the accurate representation of atomistic or continuum phenomena underlying the process [13,36,37], accurate real-time process prediction and underlying analytical process dynamics for AFSD remain outstanding issues.

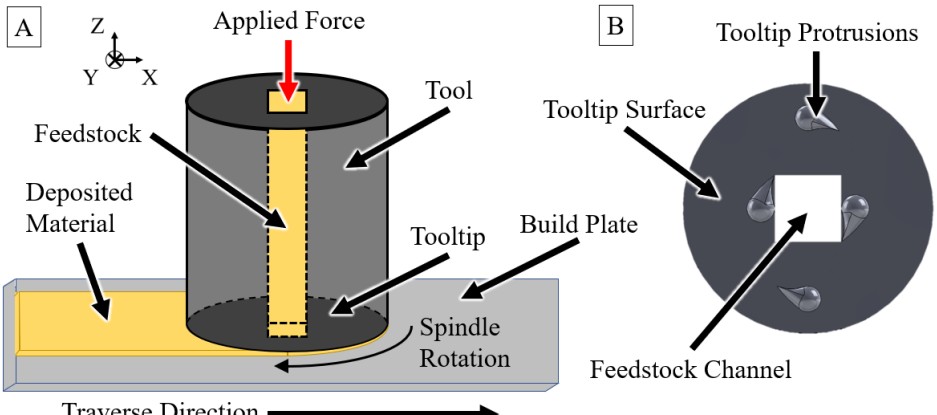

**Figure 1.** (**A**) The AFSD process. (**B**) The AFSD tool-shoulder surface.

With respect to automation, temperature control [38–40] and force control [10,41] have been successfully applied to FSW. Due to its additive nature, however, AFSD is more difficult to automate than FSW. Currently, AFSD relies on experienced technicians to monitor the build in real-time and adjust the machine (e.g., tool rotation speed or traverse rate) to ensure the weld deposition quality, material flow, and resulting microstructure. Hence, to automate AFSD, further understanding of the interactions between temperature and force control is needed. To the best of the authors' knowledge, this work represents the first published iteration of closed-loop feedback control of the AFSD process.

In this paper, closed-loop temperature and force controllers are developed to regulate the temperature at the tool-shoulder (i.e., the interaction zone) and the force applied to the feedstock in the AFSD process. Temperature control is achieved by adjusting the speed of rotation of the tool-shoulder (i.e., spindle speed), and force control is achieved by adjusting the feedrate of the consumable feedstock. The spindle speed is dictated by a three-phase alternating current (AC) motor, and the feedstock feedrate is dictated by a linear actuator that applies force at the top of the feedstock. To obtain temperature feedback, a thermocouple collar was used to wirelessly transmit temperature data from three embedded thermocouples in the tool-shoulder. Additionally, the electric current of the linear actuator was measured to obtain an estimate of the force applied to the feedstock. Experiments were conducted utilizing temperature control, force control, and combined

temperature and force control. Results are included to demonstrate the feasibility of the proposed approach.

## 2. Process Description

AFSD is governed by two interacting processes that continuously influence and depend upon each other. In the first process, the tool-shoulder rotates at the build site and stirs the substrate/feedstock materials to generate frictional heat. Protrusions, located on the working face of the tool, promote increased material mixing subsequently increasing the bond between deposited layers [42–44]. The second process involves the linear actuator that drives the consumable feedstock through the hollow tool, which is then deposited along the build path and mixed with previous layers [6,45]. An overview of the AFSD process is shown in Figure 1. Both the rotation rate of the tool (i.e., spindle speed) and the amount of force delivered by the linear actuator to the feedstock influence the amount of frictional torque generated at the interaction point and hence the temperature of the deposition. The temperature of the deposition is important because it affects the elastic modulus, shear behavior, and ductility of the feedstock. Because the spindle speed and applied feedstock force also affect the temperature of the deposition, the AFSD process can be described as dynamically cross-coupled [10]. As the deposition temperature increases and becomes more plastic, the torque required to rotate the spindle and force required to drive the feedstock into the deposition zone (i.e., interaction zone) change along with the amount of heat generated from friction. While distinctly different from AFSD, previous modeling of FSW and mesh free modeling of AFSD have suggested that the heat generated at the interaction zone is dependent on the applied force of the tool-shoulder and the rate of rotation of the spindle [13,36,46,47]. In addition to controllable inputs (i.e., spindle speed and feedrate), there are also uncontrollable factors which affect the AFSD process. For example, the build plate upon which the feedstock is deposited acts as a passive heat sink. The lack of controllable cooling suggests that the spindle speed and driving force must be carefully modulated to control the deposition temperature.

AFSD is made more complicated by the fact that many of the governing processes are nonlinear. For example, the force applied by the linear actuator to the feedstock is nonlinear because as the feedstock increases in temperature near the interaction zone, its elastic modulus and shear behavior change. Hence, as the feedstock is heated and plasticized, the amount of force at the deposition zone for a given feedrate changes. Like FSW, all the inherent process dynamics of AFSD are expected to be nonlinear and uncertain [11,46]. Additionally, there can be a delay between the application of the control effort and the subsequent change in temperature in FSW [38]. Because of the governing processes between FSW and AFSD are similar, it is assumed that there is also an input delay in AFSD. Furthermore, a variety of external factors (e.g., external cooling, ambient temperature, and material selection) can influence AFSD dynamics.

Cumulatively, AFSD is a complex nonlinear process which must be well controlled to obtain uniform builds with desirable properties. Without accurate control of AFSD, the builds are susceptible to effects of underheating (e.g., galling) or overheating (e.g., beading), as shown in Figure 2. Both of these conditions may compromise the microstructure of the build. To ensure deposition quality and avoid underheating and overheating, the build should be kept in a limited operating region around known ideal setpoints derived from evaluations of previous open loop tests (i.e., a desired temperature of 420 °C and a desired driving force of 2668 N). Setpoints, such as the desired temperature, and other properties, such as frictional force and heat capacity, change the base settings for control; thus investigations of varying control settings for different materials are warranted. For this work, we have selected aluminum alloy 6061 as the deposition material because its properties are well understood based on previous work [19,36,37].

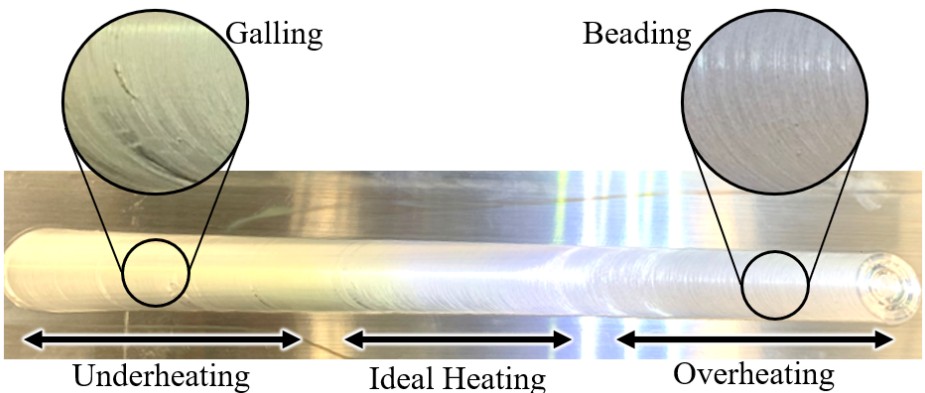

**Figure 2.** Effects of underheating, ideal heating, and overheating.

## 3. Preexisting Setup

The machine utilized in this work is a modified B8 machine from Aeroprobe. In the following, this commercially available machine (i.e., the unmodified B8) is described in terms of hardware and controllers.

### 3.1. Preexisting Hardware

The commercial machine is actuated by motors that govern the spindle speed, the linear actuator feedrate, the X-Y motion of the tabletop, and the Z motion of the tool (Figure 1). The spindle motor used to rotate the spindle and tool is a three phase AC induction motor (Baldor CEM4110T) that is controlled by an ABB motor driver (ABB ACS880-01-052A-5). The spindle has instantaneous torque, speed, and power feedback. The linear actuator used to drive the feedstock through the tool is a Kollmorgen EC5 controlled by an AKD motor driver (Kollmorgen P00607-NBEC-000). The EC5 is an electric cylinder package (EC5-AKM52G-CNR-100-10B-450-MF1-MT1E) consisting of a PMDC motor and gearing. The linear actuator has instantaneous position, speed, current, and power feedback, and an estimate of force using the current feedback. The X-Y motion of the tabletop is achieved using two ball screw actuators (one for each axis) coupled with three phase brushless PMDC motors (Kollmorgen AKM65L-ACCNR-00) that are controlled by an AKD motor driver (Kollmorgen AKD-P01207-NBEC-000). The Z motion of the tool is achieved by using an additional ball screw actuator with a gearbox and a three phase brushless PMDC motor (Kollmorgen AKM54H-ACCNR-00) and controlled by an AKD motor driver (Kollmorgen AKD-P00607-NBEC-000). All axes for translational X-Y- and Z motion have instantaneous force, position, speed, current, and power feedback.

### 3.2. Preexisting Control Design

The commercial B8 machine has controllers that regulate the spindle speed, the linear actuator feedrate, the X-Y motion of the tabletop, and the Z motion of the tool. In the default setup, the spindle and linear actuator are operated using velocity controllers, and the tabletop/tool actuators are operated using position controllers. Because the spindle motor and linear actuator are utilized to indirectly control the deposition temperature, additional details are provided.

To facilitate the subsequent control design, let $t_0 \in \mathbb{R}_{\geq 0}$ denote the initial time when the controllers are activated, and let $\zeta_d : \mathbb{R}_{\geq t_0} \to \mathbb{R}$ denote the selectable desired feedrate of the linear actuator. To control the linear actuator, a feedrate error is introduced, denoted by $\zeta_e : \mathbb{R}_{\geq t_0} \to \mathbb{R}$, defined as

$$\zeta_e(t) \triangleq \zeta_d(t) - \zeta(t), \tag{1}$$

where $\zeta : \mathbb{R}_{\geq t_0} \to \mathbb{R}$ denotes the measured feedrate using the velocity sensor on the linear actuator. The feedrate error $\zeta_e$ is utilized as the input to the preexisting velocity controller, which consists of a cascaded velocity and current control loop. The velocity controller

outputs a current command $u_\zeta : \mathbb{R} \to \mathbb{R}$ which is sent to the linear actuator, which then applies a force $F_\zeta : \mathbb{R} \to \mathbb{R}$ to drive the feedstock.

Furthermore, let $\omega_d : \mathbb{R}_{\geq t_0} \to \mathbb{R}$ denote the selectable desired spindle speed. To control the spindle, a spindle speed error is introduced, denoted by $\omega_e : \mathbb{R}_{\geq t_0} \to \mathbb{R}$ and defined as

$$\omega_e(t) \triangleq \omega_d(t) - \omega(t), \tag{2}$$

where $\omega : \mathbb{R}_{\geq t_0} \to \mathbb{R}$ denotes the measured spindle speed of the speed sensor on the spindle. The spindle speed error $\omega_e$ is utilized as the input to the preexisting speed controller, which also consists of a cascaded velocity and current control loop. The speed controller then outputs a current command $u_\omega : \mathbb{R} \to \mathbb{R}$, which is sent to the spindle's actuator, applying a torque $\tau_\omega : \mathbb{R} \to \mathbb{R}$ to rotate the spindle. The control structures for the linear actuator and the spindle are depicted in Figure 3. The X-Y motion of the tabletop and Z motion of the tool are controlled by using position controllers. The position controllers consist of cascaded position, velocity, and current control loops [48]. All controllers covered in the section are fixed in the design and are henceforth referred to as inner loop controllers.

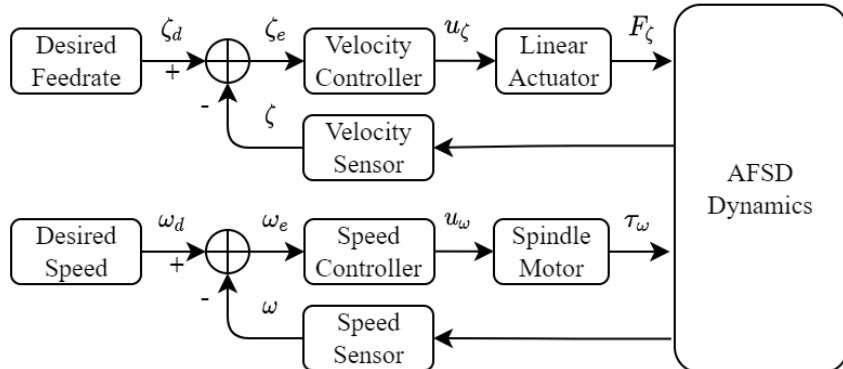

**Figure 3.** The preexisting inner loop feedrate and spindle speed controllers on the B8 machine.

## 4. Modified Setup

A thermocouple collar was developed to facilitate temperature control and obtain temperature feedback. The thermocouple collar was attached to the top of the tool and wirelessly transmitted temperature readings from three thermocouples embedded in a custom machine tool-shoulder to a base station attached to a host computer. Additionally, a custom cooling collar mount was designed to prevent the generated heat from reaching the thermocouple collar and affecting the electronics. Temperature and force controllers were then developed to close the loop using temperature and force feedback. The newly developed temperature and force controllers are referred to as outer loop controllers.

### 4.1. Modified Hardware

To obtain temperature feedback, three type-K thermocouples (Omega, Norwalk, CT, USA, TJ36-CAXL-032U-12-SB) are embedded in the custom tool, shown in Figure 4. By embedding the thermocouples within the tool, the thermocouples can pass underneath the cooling collar and obtain temperature readings at the tool-shoulder (i.e., the interaction zone). The three thermocouples are positioned at various distances from the center of the tool (i.e., 0.635 cm, 1.270 cm, and 1.905 cm) to measure an average temperature reading of the deposition. The tool is machined from high carbon tool steel and the thermocouple channels are formed using wire-drop electrical discharge machining.

The cooling collar is implemented to prevent overheating and to protect the thermocouple collar. The collar is 3d printed using photopolymer resin and designed to house three wireless thermocouple transmitters (LORD microstrain, Williston, VT, USA, TC-Link-200-OEM), a battery (Lithium Ion, 3.7 V 2200 mA), and a power distribution board (PowerBoost 1000 Charger). The temperature data are transmitted with an interval of

300 msec to a USB bluetooth base station (LORD microstrain, WSDA-BASE-101 Analog Output Base Station). The temperature-sensing module, consisting of the collar and base station is pictured in Figure 5.

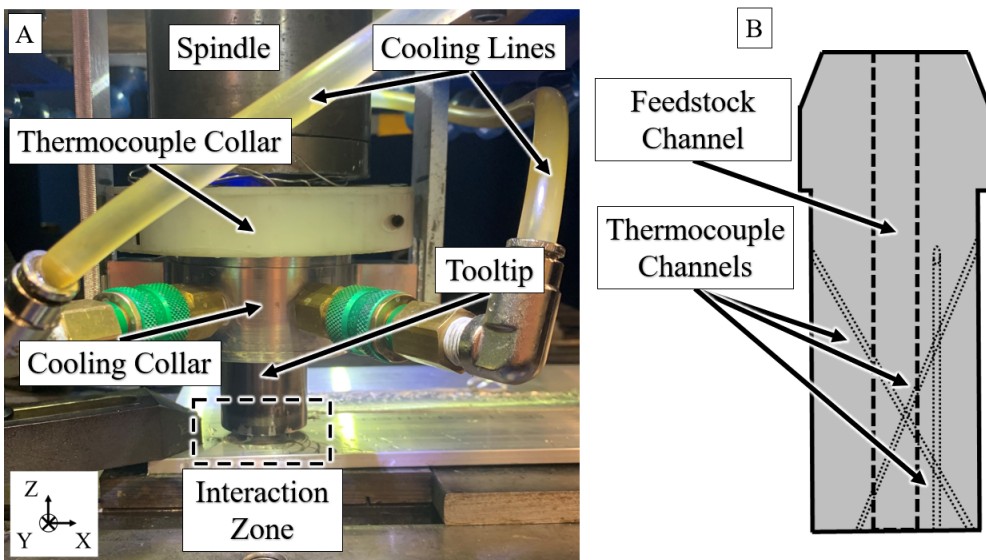

**Figure 4.** (**A**) The modified setup depicting the new thermocouple collar and modified cooling collar. (**B**) The modified tool with thermocouple channels.

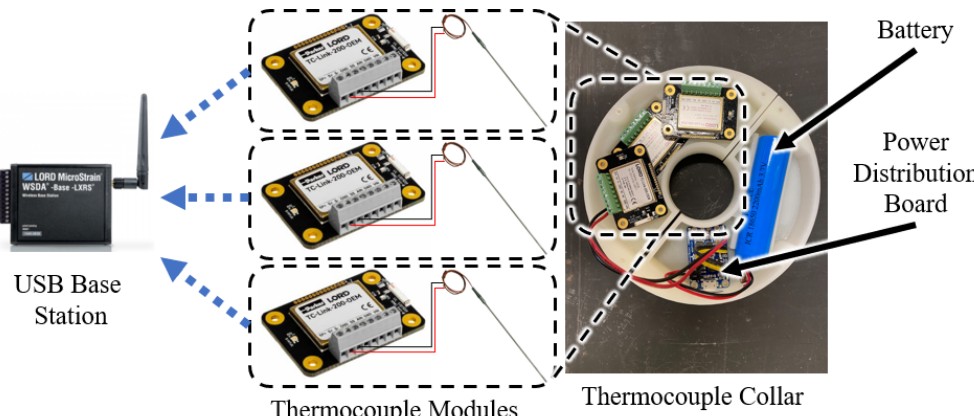

**Figure 5.** The thermocouple collar with wireless thermocouple transmitters, battery, power distribution board, and USB Base Station.

## 4.2. Modified Control Design

The control design is motivated by Fehrenbacher et al. [38] and the dynamics described in Section 2. As stated, the AFSD machine has preexisting inner loop controllers for the linear actuator and the spindle that utilize the errors in (1) and (2), respectively. These controllers remain in-place and unmodified. In the following, the linear actuator feedrate and spindle speed are utilized to indirectly control the force applied to the feedstock and the temperature interaction zone, respectively. The newly developed temperature and force controllers are outer loop controllers, built around the inner loop controllers shown in Figure 3. The combined control structure is depicted in Figure 6.

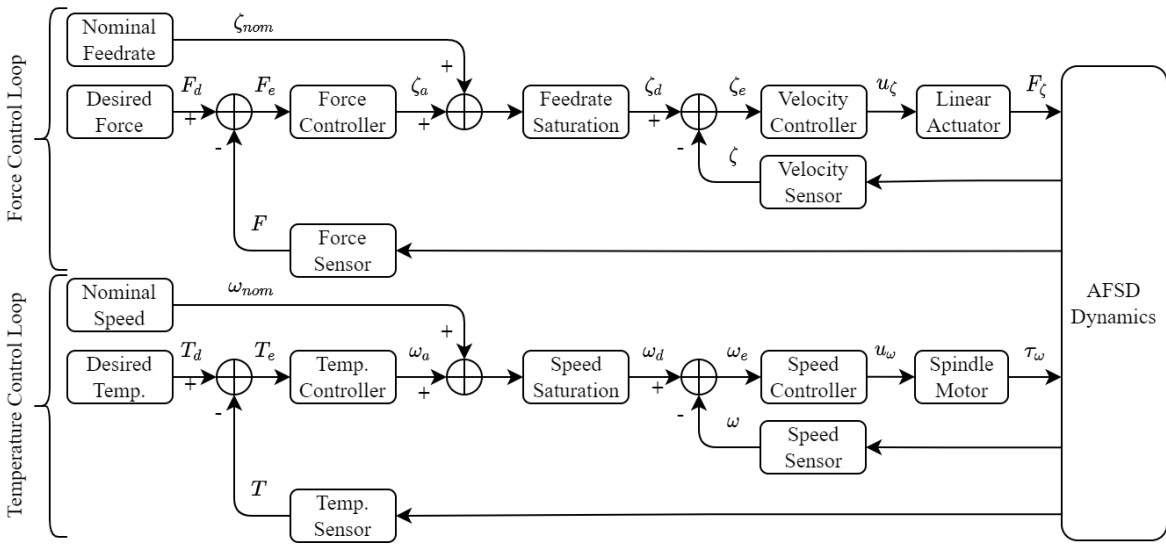

**Figure 6.** The combined control structure depicting the developed outer loop temperature and force controllers. Inputs include the nominal feedrate of the linear actuator, the nominal spindle speed, the desired temperature of the deposition, and the desired force applied to the feedstock. Temperature feedback is enabled using the developed thermocouple collar and the custom tool. Force feedback is enabled using the measured current from the linear actuator.

4.2.1. Temperature Control

To facilitate the following analysis, let $T : \mathbb{R}_{\geq t_0} \to \mathbb{R}$ denote the average measured temperature at the tool-shoulder defined as

$$T(t) \triangleq \frac{1}{3} \sum_{i=1}^{3} T_i(t), \tag{3}$$

where $T_i : \mathbb{R}_{\geq t_0} \to \mathbb{R}$ for each $i = \{1, 2, 3\}$ denotes the measured temperature of each of the three thermocouples embedded in the tool-shoulder. The temperature error is denoted by $T_e : \mathbb{R}_{\geq t_0} \to \mathbb{R}$ and defined as

$$T_e(t) \triangleq T_d(t) - T(t), \tag{4}$$

where $T_d : \mathbb{R}_{\geq t_0} \to \mathbb{R}$ denotes the selectable desired temperature at the tool-shoulder. The desired temperature can consist of a setpoint or time-varying reference trajectory. To generate an adjusted spindle speed $\omega_a : \mathbb{R} \to \mathbb{R}$, the temperature error is used as the input to the outer loop temperature controller given by

$$\omega_a(T_e) = k_{p,T} T_e(t) + k_{i,T} \int_{t_0}^{t} T_e(\tau) d\tau, \tag{5}$$

where $k_{p,T}$, $k_{i,T} \in \mathbb{R}_{\geq 0}$ denote proportional and integral gains, respectively. The new desired spindle speed $\omega_d : \mathbb{R}^2 \to \mathbb{R}$ is then calculated as

$$\omega_d(\omega_{nom}, \omega_a) = \mathrm{sat}_\omega[\omega_{nom}(t) + \omega_a(T_e)], \tag{6}$$

where $\omega_{nom} : \mathbb{R}_{\geq t_0} \to \mathbb{R}$ denotes the nominal spindle speed (i.e., the previous desired spindle speed when using the inner loop controllers). The saturation function $\mathrm{sat}_\omega[\cdot] : \mathbb{R}^2 \to \mathbb{R}$ in (6) is defined as

$$\mathrm{sat}_\omega[\omega_{nom} + \omega_a] \triangleq \begin{cases} \omega_{min} & \text{if } [\omega_{nom} + \omega_a] \leq \omega_{min}, \\ \omega_{nom} + \omega_a & \text{if } \omega_{min} < [\omega_{nom} + \omega_a] < \omega_{max}, \\ \omega_{max} & \text{if } \omega_{max} \leq [\omega_{nom} + \omega_a], \end{cases} \tag{7}$$

where $\omega_{min}$, $\omega_{max} \in \mathbb{R}_{\geq 0}$ denote the selectable minimum and maximum allowable spindle speeds, respectively. The desired spindle speed $\omega_d$ serves as the new reference for the inner loop spindle controller.

### 4.2.2. Force Control

Let $F : \mathbb{R}_{\geq 0} \to \mathbb{R}$ denote the force applied to the feedstock by the linear actuator, defined as

$$F(t) \triangleq k_f i(t), \tag{8}$$

where $k_f \in \mathbb{R}_{\geq 0}$ denotes the force constant of the linear actuator [48] and $i : \mathbb{R}_{\geq t_0} \to \mathbb{R}$ denotes the measured current of the linear actuator. In a similar manner to the temperature control scheme, the force error is denoted by $F_e : \mathbb{R}_{\geq 0} \to \mathbb{R}$ and defined as

$$F_e(t) \triangleq F_d(t) - F(t), \tag{9}$$

where $F_d : \mathbb{R}_{\geq t_0} \to \mathbb{R}$ denotes the selectable desired force applied to the feedstock by the linear actuator. To generate an adjusted feedrate of the linear actuator $\zeta_a : \mathbb{R} \to \mathbb{R}$, the force error is used as the input to the outer loop force controller given by

$$\zeta_a(F_e) = k_{p,F} F_e(t) + k_{i,F} \int_{t_0}^{t} F_e(\tau) d\tau, \tag{10}$$

where $k_{p,F}$, $k_{i,F} \in \mathbb{R}_{\geq 0}$ denote proportional and integral gains, respectively. The new desired feedrate $\zeta_d : \mathbb{R}^2 \to \mathbb{R}$ is then calculated as

$$\zeta_d(\zeta_{nom}, \zeta_a) = \text{sat}_\zeta[\zeta_{nom}(t) + \zeta_a(F_e)], \tag{11}$$

where $\zeta_{nom} : \mathbb{R}_{\geq t_0} \to \mathbb{R}$ denotes the nominal feedrate (i.e., the previous desired feedrate when using the inner loop controllers). The saturation function $\text{sat}_\zeta[\cdot] : \mathbb{R}^2 \to \mathbb{R}$ in (11) is defined as

$$\text{sat}_\zeta[\zeta_{nom} + \zeta_a] \triangleq \begin{cases} \zeta_{min} & \text{if } [\zeta_{nom} + \zeta_a] \leq \zeta_{min}, \\ \zeta_{nom} + \zeta_a & \text{if } \zeta_{min} < [\zeta_{nom} + \zeta_a] < \zeta_{max}, \\ \zeta_{max} & \text{if } \zeta_{max} \leq [\zeta_{nom} + \zeta_a], \end{cases} \tag{12}$$

where $\zeta_{min}$, $\zeta_{max} \in \mathbb{R}_{\geq 0}$ denote the selectable minimum and maximum allowable feedrates, respectively. The desired feedrate $\zeta_d$ serves as the new reference for the inner loop linear actuator controller.

## 5. Experimental Procedure and Results

Experiments were performed using aluminum alloy 6061 as the feedstock material and deposited on a plate of 6061 aluminum to demonstrate the efficacy of the temperature and force controllers. Section 5.1 provides details on the implementation of the controllers, including setpoints and controller gains. Section 5.2 details the experimental design along with a description of individual experimental protocols. Section 5.3 lists the results and provides an accompanying discussion.

### 5.1. Controller Implementation

The thermocouple collar provides updates to the host PC every 300 msec via its wireless connection. The updates consist of three thermocouple readings which are averaged according to (3) for use in feedback. The controller gains in (5) are selected as $k_{p,T} = 5.00$ and $k_{i,T} = 0.20$. The nominal filtered spindle speed in (6) is selected as $\omega_{nom} = 300$ rpm and the minimum and maximum spindle speeds in (7) are selected as $\omega_{min} = 150$ rpm and $\omega_{max} = 500$ rpm, respectively.

To obtain force feedback measurement in (8), the current measured by the Kollmorgen AKD drive is multiplied by the motor force constant of 9613 N/Ampere [48]. The sampling

interval for the measured current is 10 msec. To eliminate high frequency noise, the force measurement was filtered by combining 99% of the previously sampled value with 1% of the currently sampled value. The controller gains in (10) are selected as $k_{p,F} = 5.00 \cdot 10^{-4}$ and $k_{i,F} = 7.50 \cdot 10^{-4}$. The nominal feedrate in (11) is selected as $\zeta_{nom} = 6.99$ cm/min and the minimum and maximum feedrates in (12) are selected as $\zeta_{min} = 1.91$ cm/min and $\zeta_{max} = 10.16$ cm/min, respectively. The traverse rate was selected as 12.70 cm/min.

To prevent large control actuation, integral anti-windup is utilized on the temperature and force controllers in (5) and (10). The integral anti-windup saturates the integral terms and prevents over-accumulation of the error. All gains were initially selected and subsequently tuned for performance. If complete signal loss of the thermocouple collar occurs, the outer loop controllers are designed to deactivate.

### 5.2. Experimental Design

Experiments were divided into seven protocols with the first three protocols (i.e., Protocols A–C) utilizing force control alone. In Protocols A–C, the force controller in (10) was activated, the temperature controller in (5) was deactivated, and the nominal spindle speed $\omega_{nom}$ was used as the desired spindle speed $\omega_d$ (i.e., the input to the inner loop spindle controller). In Protocol A, the desired force $F_d$ in (9) was initially selected as 2447 N and then increased to 2669 N and 2891 N in discrete increments. In Protocol B, the desired force in (9) was initially selected as 2669 N and then decreased to 2447 N in a discrete increment. In Protocol C, a continuous time-varying tracking objective was proposed; the desired force was selected as a sinusoidally varying trajectory with an offset of 2224 N, an amplitude of 222 N, and a period of 1 min.

The next three protocols (i.e., Protocols D–F) utilized temperature control alone; in these protocols, the temperature controller in (5) was activated, the force controller in (10) was deactivated, and the nominal feedrate was used as the desired feedrate (i.e., the input to the inner loop linear actuator controller). In Protocol D, the desired temperature in (4) was initially selected as 400 °C, and then increased to 420 °C and 440 °C in discrete increments based on an estimate of settling determined by the operators. In Protocol E, the desired temperature in (4) was initially selected as 420 °C, and then decreased to 400 °C in a discrete increment based on an estimate of settling determined by the operators. In Protocol F, another continuous time-varying tracking objective was proposed; the desired temperature was selected as a sinusoidally varying trajectory with an offset of 420 °C, an amplitude of 10 °C, and a period of 2 min.

The last protocol, Protocol G, utilized combined temperature and force control with both controllers in (5) and (10) activated. For this protocol, the desired temperature and force were selected as constant setpoints of 420 °C and 2669 N, respectively.

### 5.3. Results and Discussion

A visual of the resultant weld from Protocol G is displayed in Figure 7, the tracking results of Protocols A–G are visually shown in Figures 8–14, respectively, and numerical results for all protocols are displayed in Table 1. Table 1 provides details in terms of mean tracking errors, standard deviation of the errors, and max tracking errors. For Protocols A–C (force control only), the temperature error $T_e$ is left blank; similarly, for Protocols D–F (temperature control only), the force error $F_e$ is left blank.

Across Protocols A–C, the average force tracking error was 130.8 ± 187.7 N. As observed in Figures 8–10, the measured force value is prone to noise and oscillatory behavior. Consequently, the measured force value was filtered before use in the force controller. The proportional gain in (10) assisted in improving transient performance, whereas the integral gain assisted in reducing the steady-state tracking error. Because of the integral component of (10), the controller required some time for the errors to converge to their steady-state value. During Protocol C, the desired feedrate approached the minimum $\omega_{min}$, but never reached it.

Across Protocols D–F, the average temperature tracking error was 6.9 ± 5.0 °C. As observed in Figures 11–13, the temperature controller was able to better accommodate a positive temperature error $T_e$ (i.e., when the measured temperature was below the desired temperature) than a negative temperature error. The temperature controller can increase the spindle speed and quickly generate more friction and, consequently, heat. However, due to the large heat capacity of the tool and build plate, the controller is unable to quickly and efficiently remove heat from the system and decrease the temperature of the build. As a result, the spindle speed decreases, tends toward $\omega_{min}$, and the build passively cools. This lack of active cooling in the current setup hinders the performance temperature controller and creates a unidirectional control input. Unlike the force controller, the temperature controller saturated the spindle speed at the lowest value due to the nature of the unidirectional input (see Figures 12 and 13). Hence, it is recommended that the temperature controller be tuned to avoid overshoot because it is more capable at heating the build than cooling it.

**Table 1.** Average tracking results for Protocols A–G.

| Protocol | $T_e$ (°C) | $F_e$ (N) | Max Error |
|----------|-----------|-----------|-----------|
| A | - - | 102.3 ± 130.8 | 396.8 N |
| B | - - | 151.2 ± 257.1 | 350.1 N |
| C | - - | 138.3 ± 175.3 | 565.8 N |
| D | 6.0 ± 6.8 | - - | 8.1 °C |
| E | 8.2 ± 4.5 | - - | 11.3 °C |
| F | 6.5 ± 3.8 | - - | 10.5 °C |
| G | 5.4 ± 6.5 | 140.1 ± 213.5 | 13.2 °C, 376.9 N |

Protocol G utilized combined temperature and force control. For an ambient temperature of 20 °C, the average temperature and force tracking errors were regulated to 5.4 ± 6.5 °C (1.4 ± 1.6%) and 140.1 ± 213.5 N (5.2 ± 8.0%), respectively. As seen in Table 1, the tracking errors were similar to those of temperature control and force control only. It is surmised that because both the spindle speed and the feedrate affect the temperature of the deposition, the max error of the temperature is larger than that of the previous protocols. The gains utilized in Protocol F were the same as in the previous protocols, implying that the controllers may be able to be tuned separately and then combined. Protocol F demonstrates a more traditional AFSD process, where the desired temperature and force values are constant setpoints, as opposed to time-varying trajectories.

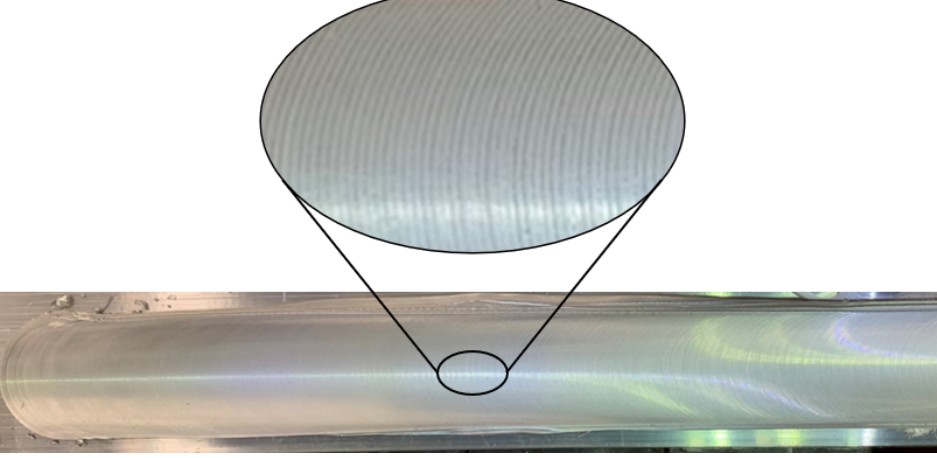

**Figure 7.** The resulting weld from Protocol G (i.e., combined temperature and force control). No galling or beading was observed.

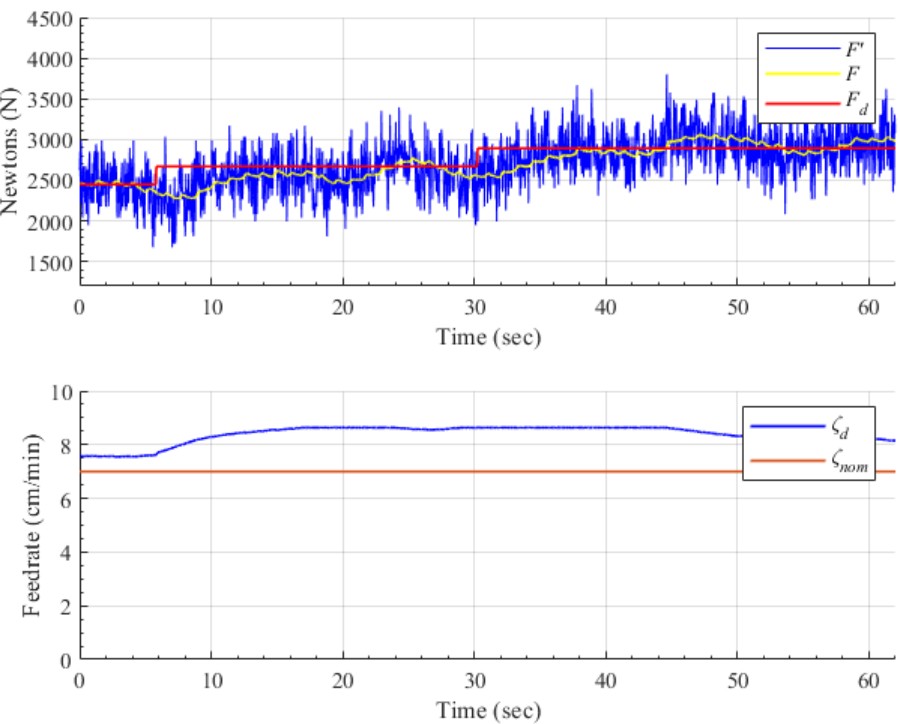

**Figure 8.** Protocol A. (**Top**) The desired versus measured force. Note, *F* denotes the filtered version of *F'*, where *F* was used in feedback. The desired force was manually increased at $t \approx 4$ s and $t \approx 29$ s. A maximum error of 396.8 N in the filtered force data occurs at $t \approx 45$ s. (**Bottom**) The desired and nominal feedrates. The desired feedrate is calculated using (8)–(12) and serves as the reference for the inner loop linear actuator controller.

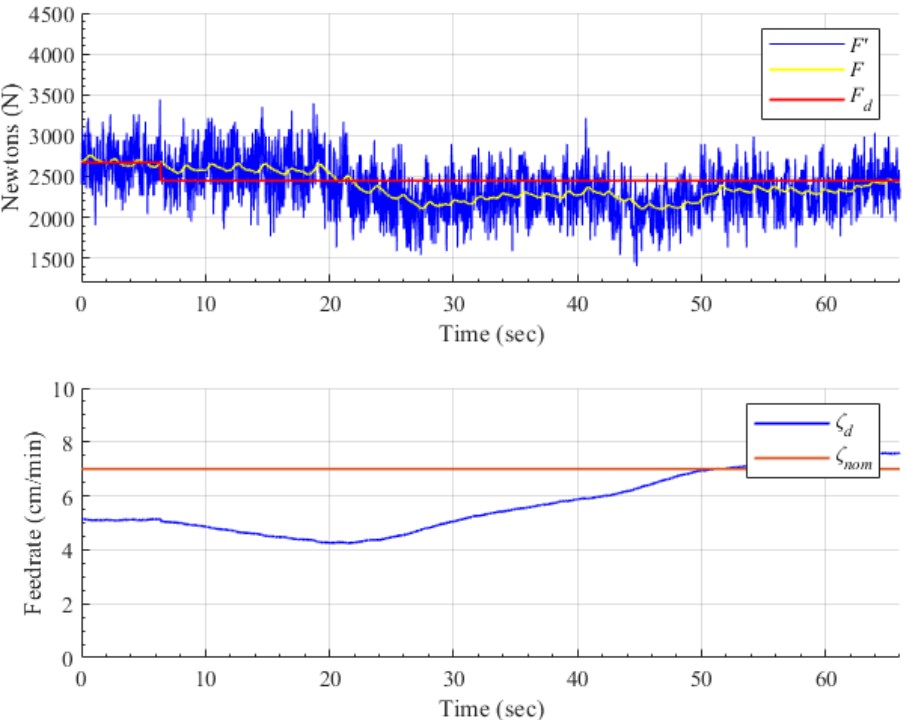

**Figure 9.** Protocol B. (**Top**) The desired versus measured force. The desired force was decreased at $t \approx 7$ s. A maximum error of 350.1 N in the filtered force data occurs at $t \approx 29$ s. (**Bottom**) The desired and nominal feedrates.

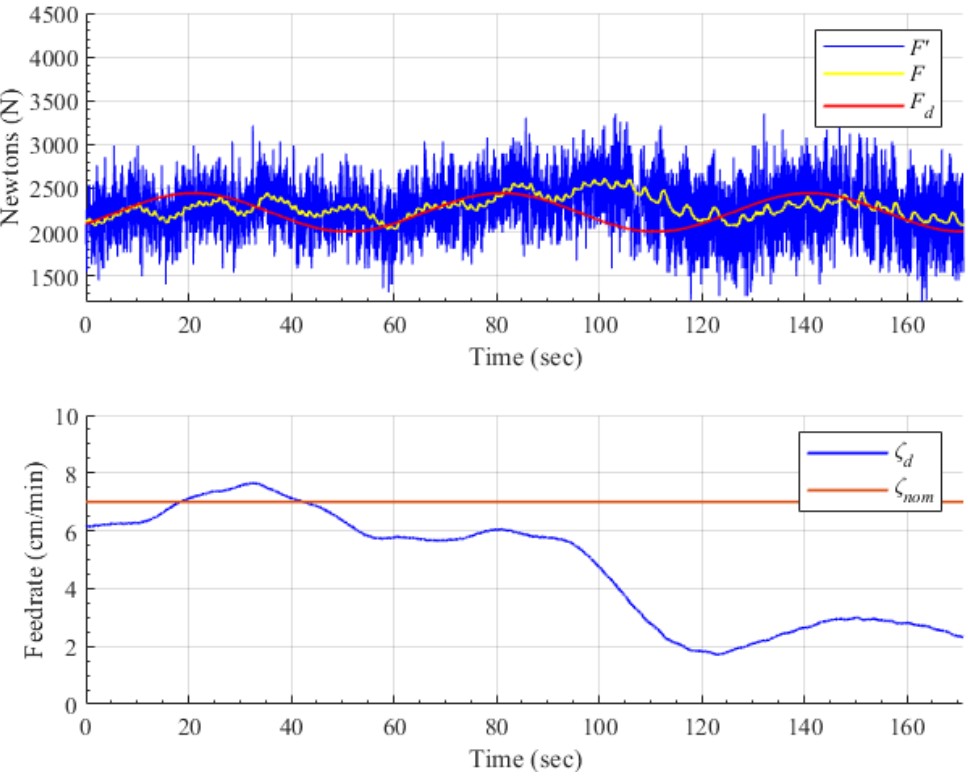

**Figure 10.** Protocol C. (**Top**) The desired versus measured force. A maximum error of 565.8 N in the filtered force data occurs at $t \approx 105$ s. (**Bottom**) The desired and nominal feedrates.

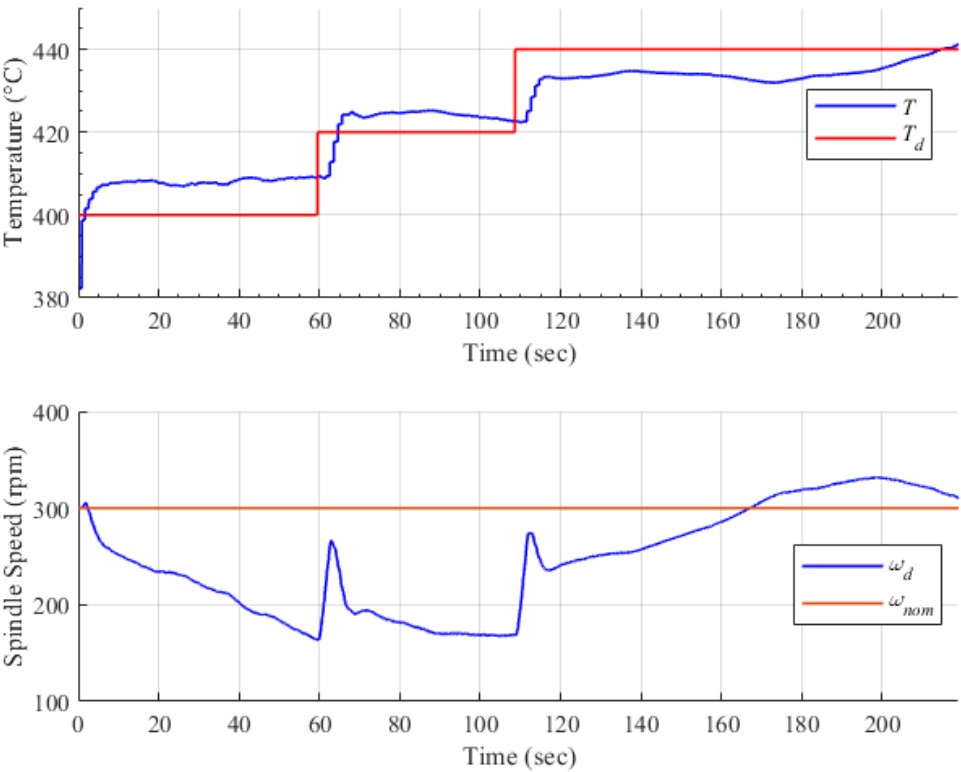

**Figure 11.** Protocol D. (**Top**) The desired versus measured temperature. The desired temperature was manually increased at $t \approx 59$ s and $t \approx 108$ s. A maximum error of 8.1 °C occurs at $t \approx 170$ s. (**Bottom**) The desired and nominal spindle speeds. The desired spindle speed is calculated using (3)–(7) and serves as the reference for the inner loop spindle controller.

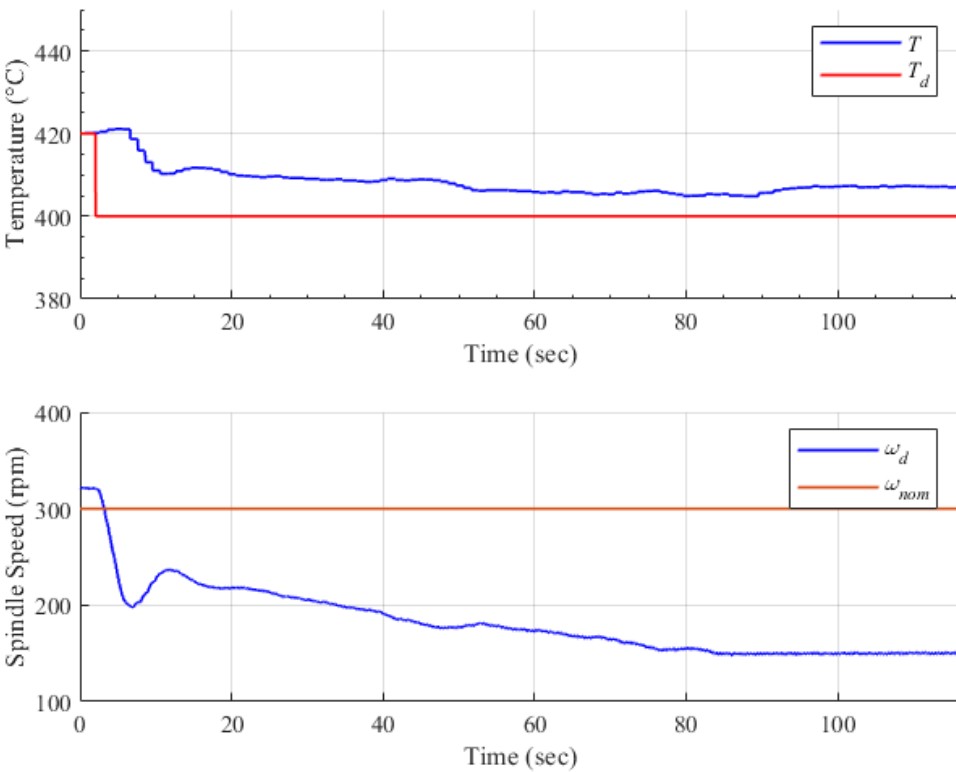

**Figure 12.** Protocol E. (**Top**) The desired versus measured temperature. The desired temperature was decreased at $t \approx 5$ s. A maximum error of 11.3 °C occurs at $t \approx 170$ s. (**Bottom**) The desired and nominal spindle speeds.

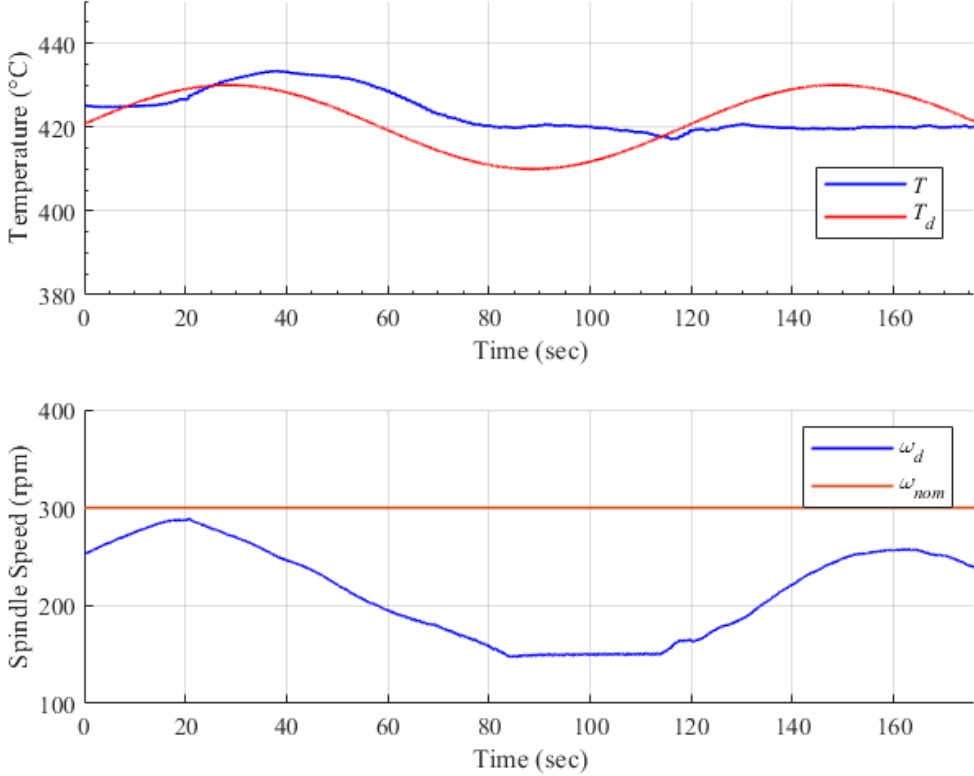

**Figure 13.** Protocol F. (**Top**) The desired versus measured temperature. A maximum error of 10.5 °C occurs at $t \approx 150$ s. (**Bottom**) The desired and nominal spindle speeds. Observe that the desired spindle speed $\omega_d$ is saturated at $\omega_{min}$ from $t \approx 85$ s to $t \approx 115$ s.

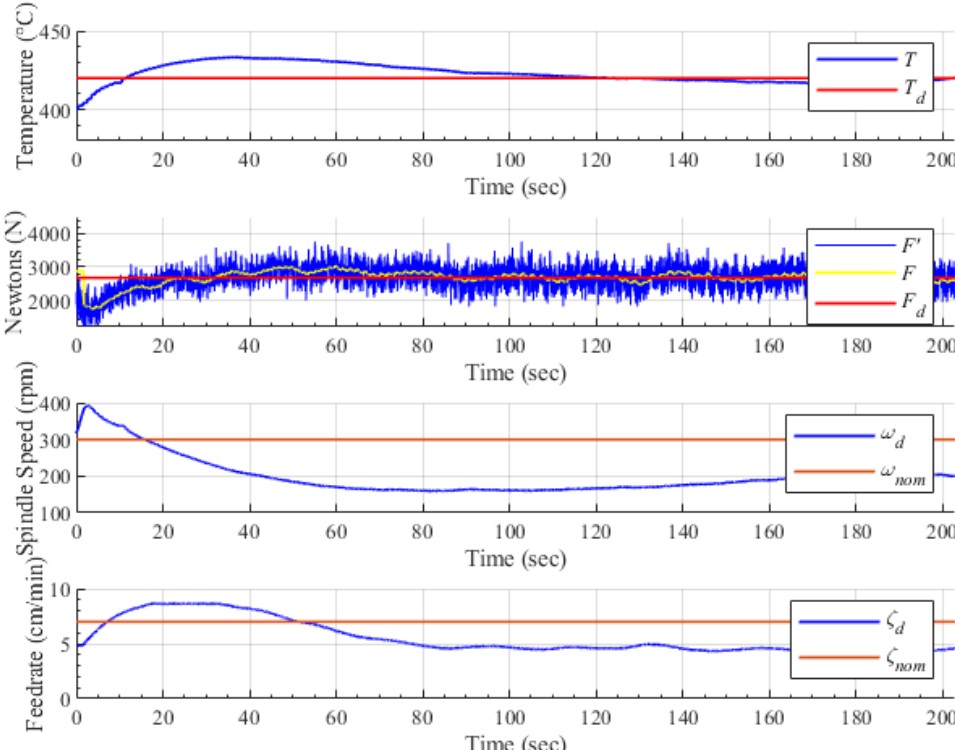

**Figure 14.** Protocol G. (**Top**) The desired versus measured temperature. A maximum error of 13.2 °C occurs at $t \approx 35$ s. (**Middle Top**) The desired versus measured force. A maximum error of 367.9 N occurs at $t \approx 45$ s. (**Middle Bottom**) The desired and nominal spindle speeds. (**Bottom**) The desired and nominal feedrates.

## 6. Conclusions

In this paper, closed-loop temperature and force control of the AFSD process was designed and implemented. Hardware modifications were made to a commercially available B8 machine to obtain temperature feedback, including the design of a new thermocouple collar and a tool to embed thermocouples at the tool-shoulder. New outer loop temperature and force controllers were developed to work in conjunction with preexisting inner loop linear actuator and spindle controllers. To demonstrate the efficacy of the proposed controllers, seven experimental protocols were performed to evaluate the temperature and force controllers individually and together. The protocols consisted of setpoint regulation and time-varying trajectory tracking. To further mature closed loop control of AFSD, future works should strive to generate dynamic models for the AFSD process. With improved models, controllers can better compensate for the dynamics present in the system, achieve smaller tracking errors, and generate improved builds with more uniform microstructures. Additionally, when the feedrate of the linear actuator deviates from the nominal feedrate, the volume of deposited feedstock deviates. Consequently, the traverse rate of the tabletop should be adjusted to preserve the volume of the deposited feedstock. Because AFSD exhibits nonlinear, uncertain, and unknown dynamics, nonlinear controllers and methods are also warranted [49]. Additionally, instead of using the force applied by the linear actuator to the feedstock for feedback, the force applied at the deposition zone can be measured directly using methods such as those used in [50,51]. Furthermore, because of the input delay present in the system [52], controllers should be designed to compensate for this delay. Lastly, future work should evaluate multi-layer builds and the quality of the build using techniques such as scanning electron microscopy.



**Author Contributions:** Conceptualization, G.R.M., P.G.A., J.B.J. and C.A.C.; Formal analysis, G.R.M. and C.A.C.; Visualization, G.R.M., M.B.W. and C.A.C.; Writing—original draft, G.R.M. and C.A.C.; Writing—review & editing, M.B.W., P.G.A., J.B.J., T.W.R. and C.A.C.; Supervision, P.G.A., J.B.J., T.W.R. and C.A.C.; Funding acquisition, P.G.A. and J.B.J. All authors have read and agreed to the published version of the manuscript.

**Funding:** The research described and the resulting data presented herein, unless otherwise noted, were funded under PE 0603119A, Project BO3 "Ground Advanced Technology/Military Engineering Technology Demonstration", Task SB0328 under Contract W912HZ1990001, Agreement W912HZ209F0004 managed by the US Army Engineer Research and Development Center. The work described in this document was conducted in the Manufacturing at the Point-of-Need Center at The University of Alabama. OPSEC permission was granted to publish this information.

**Data Availability Statement:** Not applicable.

**Acknowledgments:** The authors would like to thank Arnold Wright and Yuri Hovanski, for providing initial guidance on the design of the thermocouple collar.

**Conflicts of Interest:** The authors declare no conflict of interest. The funders had no role in the design of the study; in the collection, analyses, or interpretation of data; in the writing of the manuscript, or in the decision to publish the results.

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
