# Peer review of "Closed-Loop Temperature and Force Control of Additive Friction Stir Deposition"

_jmmp, doi:10.3390/jmmp6050092_

Round 1

Reviewer 1 Report

Review report

on the manuscript entitled “Closed-Loop Temperature and Force Control of Additive Friction Stir Deposition” by authors Glen R. Merritt, Malcolm B. Williams, Paul G. Allison, J. Brian Jordon, Timothy W. Rushing, and Christian A. Cousin (Manuscript code: JMMP-1875053)

In this work, real-time temperature and force feedback were systematically measured to optimize the additive friction stir deposition (AFSD) process of 6061 aluminum alloy. In my opinion, this is useful work, which is obviously worthy of sharing with the scientific community. In my opinion, this work is suitable for publication in its present form.

Author Response

The authors thank the reviewer for their time, feedback, and valuable comments. The reviewers insight and opinion are much appreciated.

Reviewer 2 Report

This paper is very interesting. Some points can be improved.

(1) For the introduction, the first and second paragraphs can be shortened to briefly describe FSW but try to introduce AFSD soon.

(2) The feedstock material is aluminum alloy 6061. What material is the build plate?

(3) AFSD is a non-beam-based metal additive manufacturing achieving layer-by-layer deposition without solid-to-liquid phase transformation. Do the developed temperature and force controllers here work in multi-layers also? 

Author Response

The authors thank the reviewer for their time, feedback, and valuable comments. According to the reviewers comments, the following clarifications and additions have been made.

1.) The first two paragraphs have been shortened to more quickly arrive at the ASFD process.

2.) The build plate material is also 6061 Aluminum, but due to the mechanical nature of bonding, other metallic materials can be used. The clarification is added to the experiments section.

3.) The temperature and force controllers work over multiple layers, but the transition between layers is an open loop process. We mention that future works shall seek to demonstrate the multilayer process.

Reviewer 3 Report

This paper deals with a recent innovation in non-beam-based metal additive manufacturing by Additive Friction Stir Deposition (AFSD) to avoid the solid-to-liquid phase transformation.

The mechanisms about the processes of Additive Friction Stir Deposition (AFSD) were investigated by several characterization techniques. The studies were quite systematic and the resulted were well organized by the authors. I’d like to recommend the publication of this paper in JMMP after revision.

1.       The authors should provide some results of SEM to understand the mechanisms.

2.       In Figure 9 and 10, the authors should provide more detail explain about maximum error.

3.       In Figure 11, the reasons should be provided about the change of the spindle speed at t ≈ 59 sec and t ≈ 108 sec.

Author Response

The authors thank the reviewer for their time, feedback, and valuable comments. According to the reviewers comments, the following clarifications and additions have been made.

1.) While the focus of this work was the feasibility of controlling the AFSD process, future works shall focus on the output and attempt to update control parameters based on weld quality.

2.) Clarification is added to figures 8, 9, and 10 to clarify that the maximum force error is estimated using the filtered force error.

3.) Clarification is added to elucidate that the change in control setpoint is determined by the operators, as the system is designed for a single control setpoint and must be manually changed.